# Effects of Systemic or Local Administration of Mesenchymal Stem Cells from Patients with Osteoporosis or Osteoarthritis on Femoral Fracture Healing in a Mouse Model

**DOI:** 10.3390/biom12050722

**Published:** 2022-05-19

**Authors:** Esther Laguna, María Isabel Pérez-Núñez, Álvaro del Real, Guillermo Menéndez, José A. Sáinz-Aja, Laura López-Delgado, Carolina Sañudo, Alicia Martín, Remigio Mazorra, Diego Ferreño, Belén García-Montesinos, José A. Riancho

**Affiliations:** 1Servicio de Traumatología y Cirugía Ortopédica, Hospital Universitario Marqués de Valdecilla, Instituto de Investigación Sanitaria Valdecilla (IDIVAL), Facultad de Medicina, Universidad de Cantabria, 39008 Santander, Spain; perezmi@unican.es (M.I.P.-N.); guillermo.menendez@scsalud.es (G.M.); 2Departamento de Medicina y Psiquiatría, Instituto de Investigación Sanitaria Valdecilla (IDIVAL), Facultad de Medicina, Universidad de Cantabria, 39011 Santander, Spain; delreala@unican.es; 3Laboratorio de la División de Ciencia e Ingeniería de los Materiales (LADICIM), Escuela Técnica Superior de Ingenieros de Caminos, Canales y Puertos, Universidad de Cantabria, 39005 Santander, Spain; jose.sainz-aja@unican.es (J.A.S.-A.); diego.ferreno@unican.es (D.F.); 4Servicio de Medicina Interna, Hospital Universitario Marqués de Valdecilla, Instituto de Investigación Sanitaria Valdecilla (IDIVAL), Facultad de Medicina, Universidad de Cantabria, 39011 Santander, Spain; lauralopdel@gmail.com (L.L.-D.); jose.riancho@unican.es (J.A.R.); 5Instituto de Investigación Sanitaria Valdecilla (IDIVAL), Facultad de Medicina, Universidad de Cantabria, 39011 Santander, Spain; carolinasanudo@gmail.com (C.S.); martinrebolloalicia@gmail.com (A.M.); 6Servicio de Anatomía Patológica, Hospital Universitario Marqués de Valdecilla, 39008 Santander, Spain; remigio.mazorra@scsalud.es; 7Servicio de Cirugía Maxilofacial, Hospital Universitario Marqués de Valdecilla, 39008 Santander, Spain; belenm.garciamontesinos@scsalud.es

**Keywords:** mesenchymal stem cells, bone regeneration, animal model, micro-CT, histological study

## Abstract

The purpose of this study was to analyze the regenerative capacity of mesenchymal stem cells (MSCs) in the treatment of fractures. MSCs extracted from patients with osteoporotic hip fractures or hip osteoarthritis undergoing hip replacement surgeries were cultured and injected into mice with femoral fracture. Two experimental models were established, one for the systemic administration of MSCs (*n* = 29) and another one for local administration (*n* = 30). Fracture consolidation was assessed by micro-CT and histology. The degree of radiological consolidation and corticalization was better with MSCs from osteoporosis than from osteoarthritis, being significant after systemic administration (*p* = 0.0302 consolidation; *p* = 0.0243 corticalization). The histological degree of consolidation was also better with MSCs from osteoporosis than from osteoarthritis. Differences in histological scores after systemic infusion were as follows: Allen, *p* = 0.0278; Huo, *p* = 0.3471; and Bone Bridge, *p* = 0.0935. After local administration at the fracture site, differences in histological scores were as follows: Allen, *p* = 0.0764; Huo, *p* = 0.0256; and Bone Bridge, *p* = 0.0012. As osteoporosis and control groups were similar, those differences depended on an inhibitory influence by MSCs from patients with osteoarthritis. In conclusion, we found an unexpected impairment of consolidation induced by MSCs from patients with osteoarthritis. However, MSCs from patients with osteoporosis compared favorably with cells from patients with osteoarthritis. In other words, based on this study and previous studies, MSCs from patients with osteoporosis do not appear to have worse bone-regenerating capabilities than MSCs from non-osteoporotic individuals of similar age.

## 1. Introduction

Failure of the bone healing process after fracture is estimated at 5–10%, which represents a significant clinical and economic impact [1,2].

The treatment of consolidation problems is a great challenge. Alternative therapies to the “gold standard”, which continues to be autologous iliac crest grafting, are being investigated [2,3]. Among these, cell therapies with MSCs represent an attractive option because they are the natural source of osteoblastic precursors [4,5,6].

MSCs are often isolated from the mononuclear cells present in bone marrow washings or aspirates, taking advantage of their tendency to adhere to plastic surfaces. MSCs can be maintained in culture for more than 20 divisions and expand up to 10^9^ times. However, repeated passages in culture can lead to epigenetic and phenotypic changes [7]. MSCs can impact fracture consolidation in several ways. They are able to migrate, preferentially, to skeletal lesions, releasing cytokines and growth factors that exert anabolic actions on target tissues [8,9,10,11,12]. By differentiating these cells into osteoblastic precursors and subsequently mature musculoskeletal cells, they are then able to form new bone tissue and repair skeletal defects [13,14].

The efficiency of MSCs in bone regeneration is reflected in the numerous studies and clinical trials that currently exist [15,16,17,18]. MSCs can be administered either directly into the fracture site or by systemic injection. Preliminary studies from our group suggest that MSCs from patients with osteoporotic fractures may be actively differentiated into osteoblasts in vitro, which represents an argument for considering MSCs as a promising therapeutic tool for treating consolidation disorders [19]. However, since cells from elderly patients may experience senescence-associated changes, before proposing this therapy as valid, their regenerative capacity should be tested in vivo [20,21,22].

Therefore, this study aimed to evaluate, from the radiological and histological points of view, the consolidation of an experimental fracture treated with MSCs, investigating possible differences related to either the route of administration (systemic or local) or the source of the cells (osteoporosis or osteoarthritis).

## 2. Materials and Methods

### 2.1. Isolation and Culture of Human MSCs

MSCs were obtained from the femoral heads of patients undergoing hip replacement surgery because of osteoporotic hip fracture or osteoarthritis, establishing two experimental groups. Patients with high-energy trauma, secondary osteoporosis and secondary osteoarthritis were excluded. The mean age of both patient groups was 82.65 (osteoporosis) and 71.68 (osteoarthritis). The study was approved by the Ethics Committee in Clinical Research of Cantabria and the donors gave their written informed consent.

Trabecular bone fragments from the central part of the femoral head were obtained with a trephine and washed with 50 mL of phosphate-buffered saline (PBS). The suspended cells were separated using Ficoll gradient centrifugation. Cells at the interface were cultured on polystyrene culture flasks in Dulbecco’s Modified Eagle Medium (Sigma-Aldrich, St. Louis, MO, USA). This medium was supplemented with 10% of fetal bovine serum (FBS, Sigma-Aldrich, St. Louis, MO, USA), 1% Penicillin-Streptomycin (Sigma-Aldrich, St. Louis, MO, USA) and 1% of Amphotericin A (Sigma-Aldrich, St. Louis, MO, USA). Cells from passages 2–3 were used for the experiments.

### 2.2. Experimental Animals and Surgical Procedures

Immunodeficient NOD.CB17-Prkdc scid/J mice were obtained from Jackson Laboratories (Bar Harbor, ME, USA). The colony was housed at the animal housing and experimentation service of the University of Cantabria, under aseptic conditions and veterinary control. The animal experimental protocol was approved by the Research Ethics Committee of the University and the Health Council of Cantabria, as established by current regulations. Surgery was performed on 8-week-old mice, weighing 24–48 g, after intraperitoneal anesthesia and antibiotic. Through a lateral approach to the femur, a complete open osteotomy was performed at the level of the middle third of the right femur by cutting with a number 15 scalpel. The fracture was stabilized by retrograde nailing from the knee (intercondylar region) with a 27 G needle. Subsequently, the periosteum was cauterized at 800 °C with a portable bipolar electrocautery to slow down the bone healing process. Sedation was reversed and postoperative analgesia was administered. The animals were sacrificed 4 weeks after surgery to extract the femurs under study (Figure 1).

### 2.3. Experimental Groups

Mice were randomly divided into two experimental models: model A, for systemic administration of MSCs (*n* = 29); and model B, for local administration of MSCs (*n* = 30). In both groups, cells were injected through a 25 G needle and cells from a single patient were infused into each mouse.

Model A was further divided into group A1 (control group, *n* = 13) injected with 0.1 mL of cell-free saline; group A2 (osteoporosis group, *n* = 8) injected with 10^6^ MSCs resuspended in 0.1 mL of saline, cultured from patients with osteoporosis; and group A3 (osteoarthritis group, *n* = 8) injected with 10^6^ MSCs resuspended in 0.1 mL of saline, cultured from patients with osteoarthritis. MSCs were injected retro-orbitally on postoperative day 2.

Model B was divided into group B1 (control group, *n* = 11) injected with 0.1 mL of cell-free saline; group B2 (osteoporosis group, *n* = 9) injected with 10^6^ MSCs resuspended in 0.1 mL of saline, cultured from patients with osteoporosis; and group B3 (osteoarthritis group, *n* = 10) injected with 10^6^ MSCs resuspended in 0.1 mL of saline, cultured from patients with osteoarthritis. Cells were applied to the fracture site during surgery, with the addition of a collagen patch (Lyostypt^®^) to prevent cell migration.

### 2.4. Radiological Analysis

The femurs were scanned with the high-resolution micro-CT scanner Skyscan1172 (Bruker-microCT, Kontich, Belgium) with a resolution of 6 μm. A 2D analysis of the callus was performed with the CTAn^®^ program (v.1.18) to obtain the BV/TV (Bone Volume/Tissue Volume) ratio, which is proportional to the bone mass. We obtained two data: BV/TV callus + cortical (which refers to the complete callus, including the primary cortex) and BV/TV callus (from which the primary cortex is subtracted). The CTVox^®^ program (v.1.5) performed a 3D reconstruction, from which we selected orthogonal-coronal and sagittal views crossing at the center of the bone, to assess the four cortices around the bony callus. The assessment was carried out by four independent observers who were unaware of the treatment group. In case of discrepant scores, images were reviewed until a consensus was reached. Consolidation and corticalization of the fracture callus were assessed. To score consolidation, one point was awarded for each cortex showing a Bone Bridge between fracture edges (0–4). Callus corticalization was defined by the presence of a well-defined layer of cortical bone at the outer limit of the callus, and one point was given for each consolidated cortex at the orthogonal views of the callus (0–4).

### 2.5. Histological Analysis

After fixation in formalin, femurs were preserved in ethanol and decalcified. Then, the intramedullary needles were removed, and bone was embedded in paraffin blocks. Sections of 3 µm were cut and stained with hematoxylin–eosin.

Histological sections were evaluated using two semi-quantitative numerical scales, proposed by Allen and Huo [23,24]. In addition, the presence or absence of bone bridging in the two cortices observed has been considered (Table 1).

### 2.6. Data Analysis

Kruskal–Wallis and Mann–Whitney U tests were used for the statistical comparison of between-group differences. Non-parametric tests were used because the sample size was small, and we were not able to ascertain the distribution of the data. *p*-values < 0.05 were considered statistically significant.

## 3. Results

### 3.1. Radiological Study

When analyzing the BV/TV we did not find differences between subgroups 1 (control), 2 (osteoporosis) or 3 (osteoarthritis), within each administration group, systemic (A) or local (B). However, all parameters (callus + cortical BV/TV and callus only BV/TV) were higher in the local administration group (*p* < 0.05). When callus + cortical BV/TV callus was studied, the between-group pairwise comparisons were as follows: A1-B1—*p* = 0.0011; A2-B2—*p* = 0.0069; and A3-B3 *p* = 0.0223. When callus only BV/TV was analyzed, the p-values of the comparisons were as follows: A1-B1—*p* = 0.0001; A2-B2—*p* = 0.0069; and A3-B3—*p* = 0.0157 (Figure 2).

The analysis of consolidation and corticalization of model A (systemic administration) were as follows. Consolidation [median (IQR)]: control—4.0 (3.0–4.0); osteoporosis—3.5 (2.3–4.0); and osteoarthritis 0.5 (0–2.3). Corticalization: control—2.0 (1.0–3.0); osteoporosis—1.5 (0–2.0); and osteoarthritis—0 (0–0.3). The scores for MSCs grown from patients with osteoporosis were significantly better than those for MSCs from patients with osteoarthritis (*p* = 0.0302 consolidation; *p* = 0.0243 corticalization). The comparisons of osteoporosis and control groups did not show significant differences (Figure 3).

Consolidation scores in model B (local administration) were: control—4.0 (3.0–4.0); osteoporosis—4.0 (3.0–4.0); and osteoarthritis—3.0 (2.3–3.8). The corticalization scores were: control—2.0 (1.0–2.0); osteoporosis—1.0 (0.3–1.8); and osteoarthritis—1.0 (0.3–1.8). Although the scores tended to be better in the MSCs from patients with osteoporosis than osteoarthritis, the differences did not reach statistical significance (*p* = 0.1771 consolidation; *p* = 0.2503 corticalization). Likewise, there were no significant differences between the treatment and control groups (Figure 4).

As expected, the overall comparisons of Models A and B showed no significant differences in control subgroups that did not receive MSCs either by the systemic or the local routes. On the contrary, in animals who were transplanted with MSCs, those receiving MSCs by the systemic route tended to have somewhat lower (albeit not statistically significant) consolidation scores than those receiving MSCs locally, particularly in the case of MSCs grown from patients with osteoarthritis (*p* = 0.0993–0.6977).

The next figure shows several examples of micro-CT studies in different experimental groups (Figure 5).

### 3.2. Fracture Histology

In the systemic administration group (Model A), the histological analysis (Allen, Huo and Bone Bridge scores) showed better scores in animals receiving MSCs from osteoporosis than in those injected with MSCs from patients with osteoarthritis; this difference was significant regarding Allen (*p* = 0.0278) and Huo (*p* = 0.0347) scores, but not in Bone Bridge (*p* = 0.0935) scores. We found no differences between the control group and the osteoporosis group (Figure 6).

In the local administration group (Model B), the analysis of histological results also showed better scores with MSCs from osteoporosis than MSCs from osteoarthritis, although this improvement was not so striking. The difference was significant in the Huo (*p* = 0.0256) and Bone Bridge (*p* = 0.0012) scores, but not in the Allen (*p* = 0.0764) score. We found no differences between the control group and the osteoporosis group (Figure 7).

In the pairwise analysis comparing systemic and local administration of MSCs, we found no differences in any group (control, osteoporosis, osteoarthritis) in any of the three scales (Allen, Huo, and Bone Bridge) (*p* = 0.0829–0.8409).

The next figure shows several examples of histological studies in different experimental groups (Figure 8).

### 3.3. Correlation between Radiological and Histological Study

Overall, radiological and histological scores were strongly correlated, as shown in Figure 9.

## 4. Discussion

As natural precursors of osteoblasts, MSCs are attractive candidates to be used in bone regeneration therapies, including the filling of bone defects, and delayed fracture consolidation. Their potential role in generalized bone disorders, such as osteoporosis, has also been postulated [25].

Autologous MSCs would be particularly attractive due to the absence of rejection and other practical and ethical problems. However, the osteogenic capacity of MSCs from osteoporosis patients and other elderly subjects has been questioned. Some studies found a decreased number of circulating osteogenic precursors and decreased activity of MSCs in osteoporotic patients and elderly people. However, others have pointed out that the mobilization and activity of MSCs is preserved in the elderly [26]. In particular, the state of senescence of the MSCs of these patients is of concern. This is emerging as an important mechanism in the pathogenesis of several age-related disorders, including osteoporosis [20,22]. In previous studies, we found that MSCs from osteoporotic patients retain the ability to undergo osteogenic differentiation and express high levels of collagen and other matrix constituents in vitro. However, alkaline phosphatase activity and calcium deposition are decreased in comparison with MSCs grown from patients with osteoarthritis of a similar age range [19]. Likewise, cells from osteoporotic patients and osteoarthritic patients have a similar capacity to form new bone when transplanted subcutaneously into immunodeficient mice [18].

In this study, we aimed to explore if MSC patients undergoing hip replacement improved fracture healing in a model of suboptimal fracture consolidation by using unstable endomedullary osteosynthesis. In this model, we were not able to show any beneficial effects of MSCs from patients with osteoporosis administered either intravenously or locally at the fracture site, and the radiological and histological scores of fracture consolidation were similar in control animals and in those receiving MSCs from osteoporotic patients. On the contrary, and quite unexpectedly, animals infused with MSCs from patients with osteoarthritis showed an impairment of fracture healing scores, particularly if cells were injected intravenously.

Either by a systemic or local administration, MSCs arguably have two possible mechanisms of action. First, a direct effect using the differentiation of MSCs into osteoblasts and chondroblasts that could directly contribute to bone repair. Second, indirectly, by modulating inflammation and immune responses that participate importantly in the healing process [27,28]. In fact, MSCs release a variety of molecules, including cytokines, growth factors and miRNAs, some of them enclosed within exosomes and other extracellular vesicles, which may have paracrine and systemic regulatory effects [15]. Most MSCs injected intravenously appear to be trapped in visceral tissues, and particularly in the lungs [29,30,31]. This was also the case in this study (data not shown). Thus, the direct differentiation of MSCs into skeletal progenitor cells likely has little importance when MSCs are infused by this route.

The mechanisms explaining the impairment of fracture consolidation induced by MSCs from osteoarthritic patients are unclear. Nevertheless, since the effect was more marked after intravenous infusion than after local administration into the fracture site, they likely involve secondary effects mediated through the release of circulating factors. There are complex interaction networks between immune and MSCs cells. As recently reviewed by Chen et al. [32], innate immune responses provide a suitable inflammatory microenvironment for initiating fracture repair. Adaptive immune responses maintain bone regeneration and bone remodeling. MSCs and immune cells regulate each other. All kinds of immune cells and secreted cytokines can regulate the migration, proliferation and osteogenic differentiation of MSCs, which have a strong immunomodulatory ability to these immune cells. Additionally, T-cell deficiency may impair bone regeneration.

This study has several limitations. Most importantly, we used immunodeficient animals to prevent the rejection of human MSCs. As previously mentioned, cells of the immune system play an important role in fracture healing and modulate the activity of MSCs. Therefore, it is unclear if our results can be extrapolated to organisms with preserved immune responses, including patients with delayed union and non-union fractures. Due to ethical and practical reasons, we did not use MSCs from normal individuals, but compared cells from patients with osteoporosis and osteoarthritis, which tend to show bone mass changes in the opposite direction. However, this prevented us from comparing the effect of these MSCs with those of normal individuals. Additionally, following requirements to limit the number of animals used for experiments, the sample size of each group was relatively limited. This compromised the power in statistical comparisons, particularly in the context of wide interindividual variance of consolidation scores.

## 5. Conclusions

In our study, we were not able to demonstrate an MSCs-mediated improvement in bone healing, irrespective of the MSCs’ origin. Further experiments are needed to establish whether this reflects a suboptimal fracture fixation and other drawbacks of the experimental model, or it rather reflects a true lack of effect of MSCs. Indeed, we found an unexpected impairment of consolidation induced by MSCs from patients with osteoarthritis, but the mechanisms involved remain to be elucidated. In any case, fracture healing with MSCs from patients with osteoporosis compared favorably with cells from patients with osteoarthritis. In other words, based on this and previous studies, MSCs from patients with osteoporosis do not appear to have worse bone regenerating capabilities than MSCs from non-osteoporotic individuals of similar age.

## Figures and Tables

**Figure 1 biomolecules-12-00722-f001:**
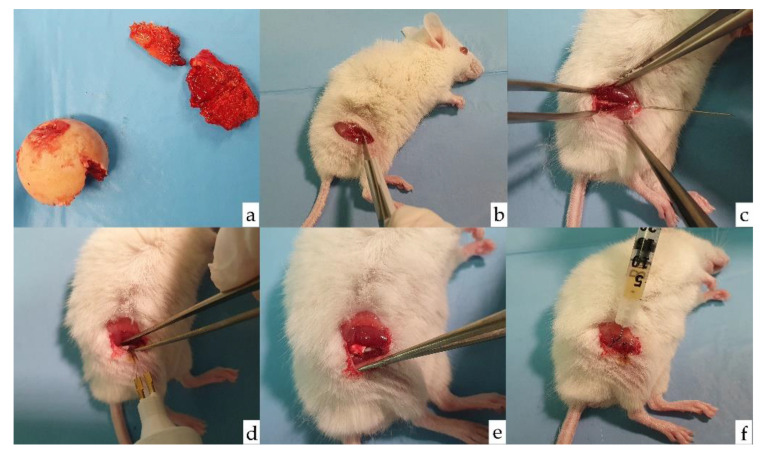
Surgical procedure. (**a**) Femoral head and neck of prosthetic replacement surgery that serves for extraction of bone cylinders and culture of MSCs. (**b**) Transverse osteotomy of the femur. (**c**) Retrograde 27 G needle insertion. (**d**) Cauterization of the fracture focus at periosteum level. (**e**) Addition of a collagen patch (Lyostypt^®^, Braun-Aesculap, Tuttlingen, Germany to prevent cell migration. (**f**) Local injection of MSCs in Model B.

**Figure 2 biomolecules-12-00722-f002:**
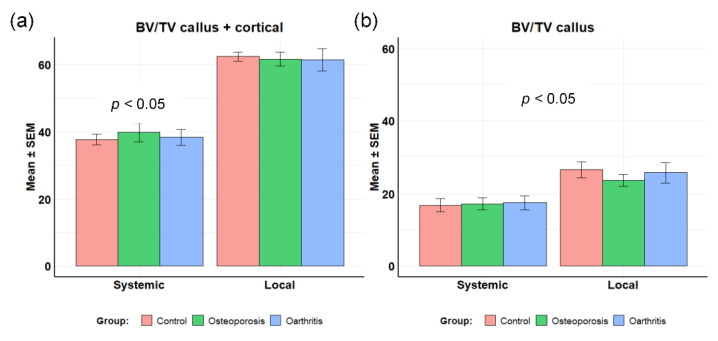
BV/TV of the callus region. (**a**) Callus+cortical BV/TV; (**b**) callus only BV/TV.

**Figure 3 biomolecules-12-00722-f003:**
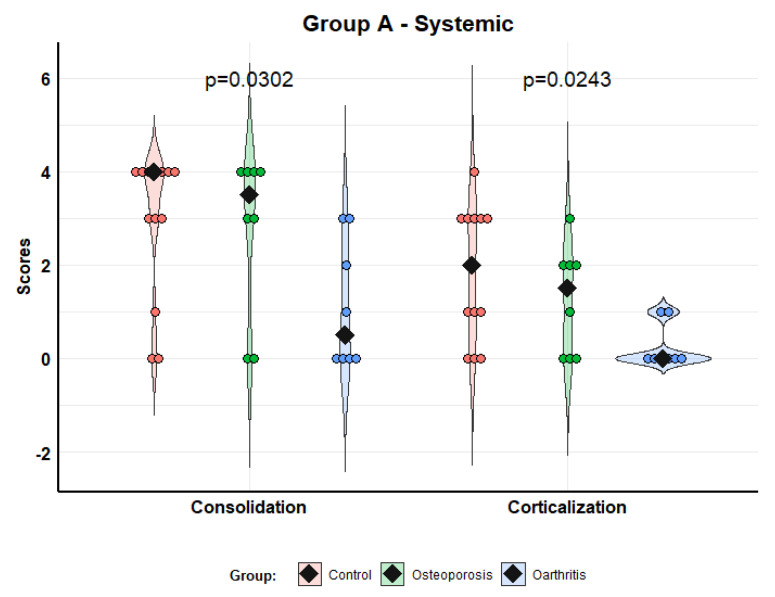
Scores of consolidation and corticalization in model A (systemic administration). Violin plots with individual values and medians (rhombi).

**Figure 4 biomolecules-12-00722-f004:**
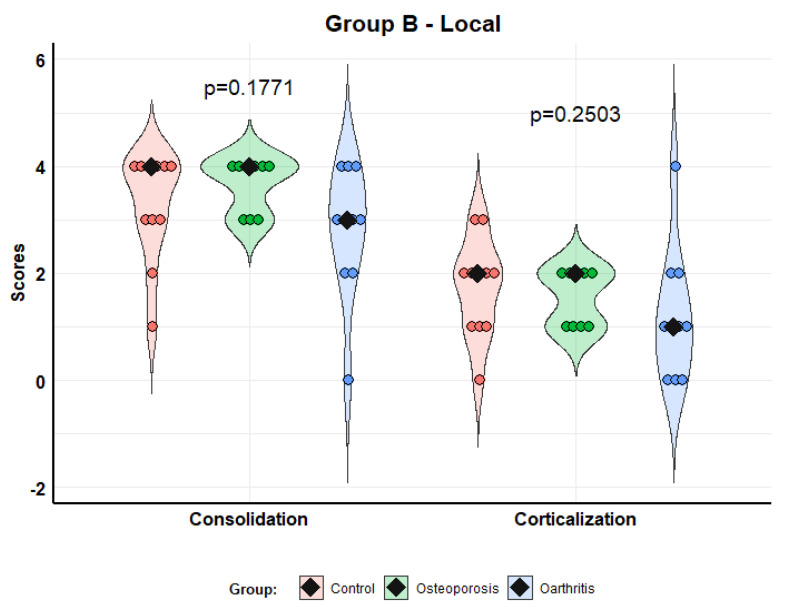
Scores of consolidation and corticalization in model B (local administration). Violin plots with individual values and medians (rhombi).

**Figure 5 biomolecules-12-00722-f005:**
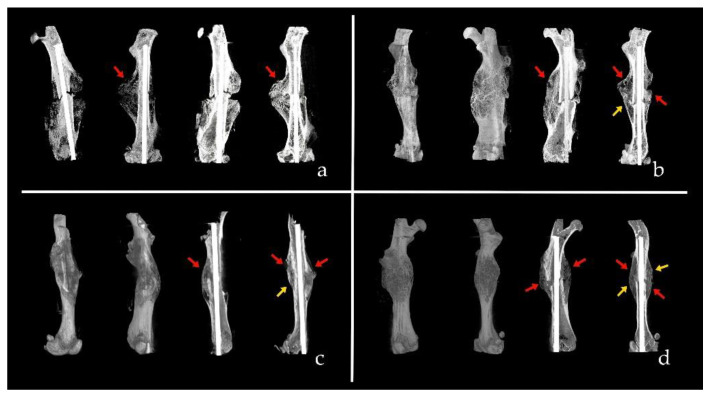
Several examples of consolidation and corticalization scores. (**a**) CT of group A3 (systemic administration of MSCs from osteoarthritis). Score 1-0: One cortex is consolidated and no one corticalized. (**b**) CT or group A2 (systemic administration of MSCs from osteoporosis). Score 3-1: Three consolidated cortices and one corticalized. (**c**) CT of group B3 (local administration of MSCs from osteoarthritis). Score 3-1: Three consolidated cortices and one corticalized. (**d**) CT of group B2 (local administration of MSCs from osteoporosis). Score 4-2: Four consolidated cortices and two corticalized. Red arrows are consolidated cortices and yellow arrows are corticalized cortices.

**Figure 6 biomolecules-12-00722-f006:**
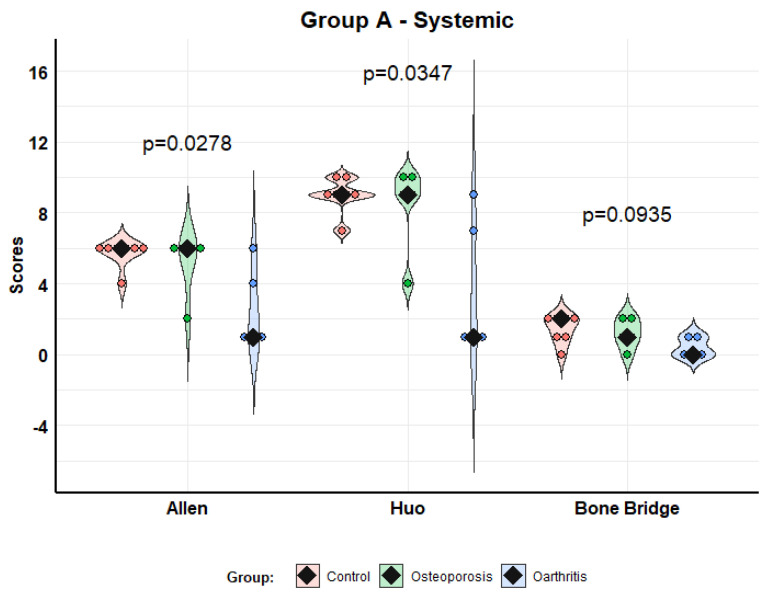
Histological scores according to Allen, Huo and Bone Bridge scales for Model A (systemic administration). Violin plots with individual values and medians (rhombi).

**Figure 7 biomolecules-12-00722-f007:**
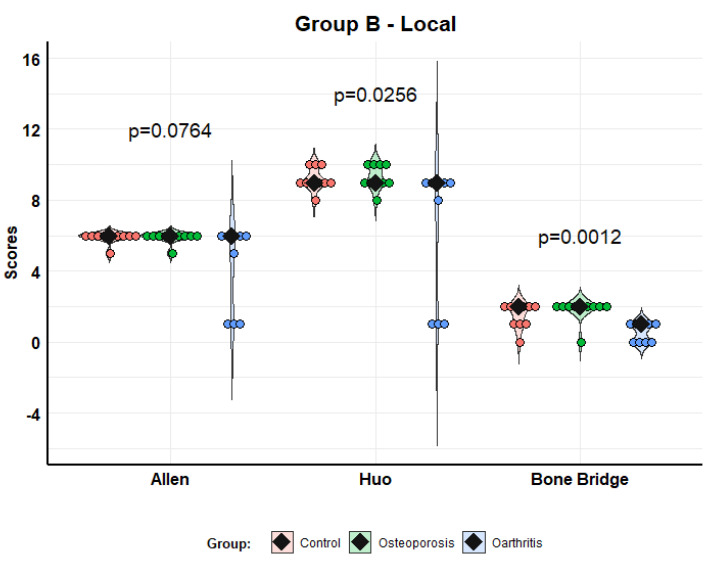
Histological scores according to Allen, Huo and Bone Bridge scales for Model B (local administration). Violin plots with individual values and medians (rhombi).

**Figure 8 biomolecules-12-00722-f008:**
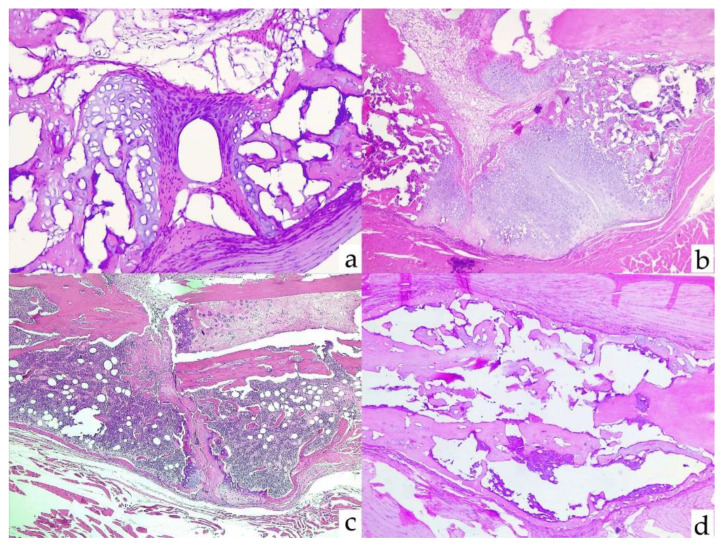
Several examples of histological scores. (**a**) Group A3: absence of consolidation. Allen 1, Huo 1, and Bone Bridge 0. (**b**) Group A2: incomplete consolidation with cartilage between bone fragments. Allen 2, Huo 4, and Bone Bridge 0. (**c**) Group B3: immature bone and cartilage. Allen 5, Huo 8, and Bone Bridge 0. (**d**) Group B2: complete consolidation with bilateral bone bridge. Allen 6, Huo 10, and Bone Bridge 2.

**Figure 9 biomolecules-12-00722-f009:**
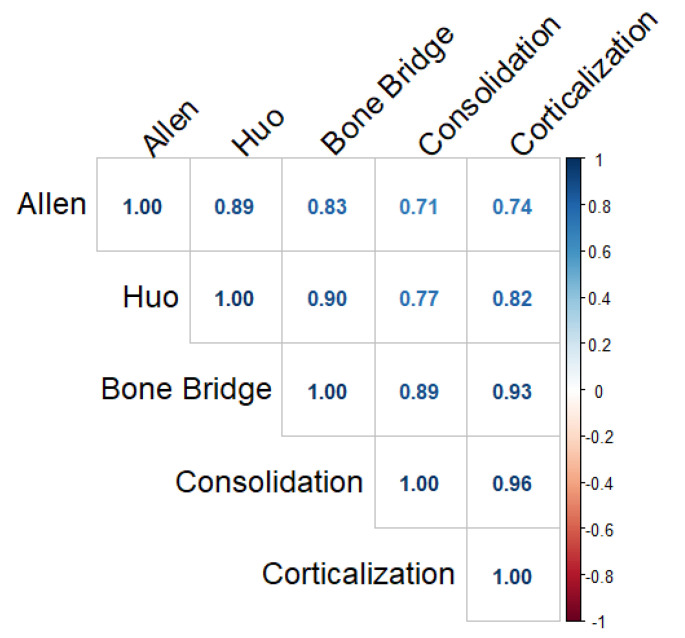
Spearman correlation coefficients between radiological and histological indices.

**Table 1 biomolecules-12-00722-t001:** Histological scores. Allen score, Huo score and Bone Bridge score to assess histological consolidation.

ALLEN	
**Score**	**Histological Findings**
**1**	Pseudoarthrosis: fibrous tissue between the fragments
**2**	Incomplete union: fibrous and cartilaginous tissue
**3**	Complete union with cartilaginous tissue
**4**	Incomplete bone union with areas of ossification: the equivalent amount of cartilage and trabecular bone
**5**	Incomplete bone union with a predominance of bone cells
**6**	Complete bone consolidation: bone between both bone fragments
**HUO**	
**Score**	**Histological findings**
**1**	Fibrous tissue
**2**	Predominantly fibrous tissue with a small amount of cartilage
**3**	An equal mixture of fibrous and cartilaginous tissue
**4**	Predominantly cartilage with a small amount of fibrous tissue
**5**	Cartilage
**6**	Predominantly cartilage with a small amount of immature bone
**7**	An equal mixture of cartilage and immature bone
**8**	Predominantly immature bone with a small amount of cartilage
**9**	Union of fracture fragments by immature bone
**10**	Union of fracture fragments by mature bone
**BONE BRIDGE**	
**Score**	**Histological findings**
**0**	No bone bridge
**1**	Bone bridge in one cortex
**2**	Bone bridge in two cortices

## Data Availability

Raw data are available from authors upon reasonable request.

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
