# Peer review of "Effects of Systemic or Local Administration of Mesenchymal Stem Cells from Patients with Osteoporosis or Osteoarthritis on Femoral Fracture Healing in a Mouse Model"

_biomolecules, 2022, doi:10.3390/biom12050722_

Round 1

Reviewer 1 Report

The manuscript assess the regeneration capacities of human BMSCs from osteoporosis and osteoarthritis patients on fracture healing of mice. The manuscript present new findings between osteoporosis and osteoarthritis human BMSCs. But some viewpoints and documentation needed to be revised and clarified.

  1. Title, the title should be revised as" Effects of systemic or local administration of mesenchymal stem cells from patients with osteoporosis or osteoarthritis on femoral fracture healing in a mice model" to let readers more easily understood.
  2. introduction, more references related to efficiency of MSCs in bone regeneration are strongly suggested, for example "Stem Cell Res Ther. 2022 Feb 19;13(1):73.; Connect Tissue Res. 2022 May;63(3):243-255.......etc "
  3. results, some results and figures related to MSCs charateristics between osteoporosis and osteoarthritis are suggested.
  4. results, the figures among the six experimental groups are needed, especially the histological findings which is very important for confirmation of the manuscript.
  5. Why selecting the MSCs from osteoporosis and osteoarthritis as experimental groups? This is because the vitality and stemness capacities of MSCs from the patients with disease is lower than those in healthy patients.
  6.  Moreover, the efficiency of MSCs among healthy, osteoporosis and osteoarthritis are similar. How to explain the findings? 
  7.  There is also lack of approval for animal experiment.        

Author Response

Point 1: Title, the title should be revised as" Effects of systemic or local administration of mesenchymal stem cells from patients with osteoporosis or osteoarthritis on femoral fracture healing in a mice model" to let readers more easily understood.

Response 1: Title has been corrected.

Point 2: Introduction, more references related to efficiency of MSCs in bone regeneration are strongly suggested, for example "Stem Cell Res Ther. 2022 Feb 19;13(1):73.; Connect Tissue Res. 2022 May;63(3):243-255.......etc "

Response 2: We have added more references, in fact one of your suggestions is for our work group and appear in the Discussion, as well.

Point 3: Results, some results and figures related to MSCs charateristics between osteoporosis and osteoarthritis are suggested.

Response 3: Figure showing results in different experimental groups related to micro-CT has been changed to the “results section”.

Point 4: Results, the figures among the six experimental groups are needed, especially the histological findings which is very important for confirmation of the manuscript.

Response 4: Figure showing results in different experimental groups related to histological study has been changed to the “results section”. I think is the correct place, instead of “Material and methods section”.

Point 5: Why selecting the MSCs from osteoporosis and osteoarthritis as experimental groups? This is because the vitality and stemness capacities of MSCs from the patients with disease is lower than those in healthy patients.

Response 5:

It is difficult to justify harvesting bone MSCs from healthy donors, for ethical an practical reasons, so we obtain the cells from femoral heads, specifically of femoral heads of patients undergoing hip replacement surgery because of two pathologies,  osteoporotic hip fracture or osteoarthritis. In the begining, we had supposed that MSCs from osteoarthritis should behave as healthy MSCs and we wanted to know what was going to happen with MSCs from osteoporosis. In that case, is the regenerative capacity diminished or not in these cells? Can we compare them with non ostoporotic MSCs?

Point 6: Moreover, the efficiency of MSCs among healthy, osteoporosis and osteoarthritis are similar. How to explain the findings?

Response 6: Unexpectedly, what we have found that MSCs from osteoarthritis show an impairmet of consolidation compared to control group (no cells) and this is more evident in the systemic administration. The mechanism is unclear. We used immunodeficient animals, which does not occur under normal conditions in humans, and the immune system play an important role in fracture healing and can modulate MSCs. It could be one of the explanations, as well as one limitation of this study because it is unclear if the results can be extrapolated to normal individuals (with healthy immune response).

On the other hand, MSCs from osteoporosis compared favorably with cells from osteoarthritis, nevertheless scores of fracture consolidation were similar in control group. So, we interpret that MSCs from osteoporotic fractures would likely be similarly valid to healthy MSCs, because its regenerative capacity is not compromised.

 Point 7: There is also lack of approval for animal experiment.  

Response 7: The IRB approval for animal and human experiments is now included in the IRB statement at the end of the text.

Thank you for your comments and suggestions.

Reviewer 2 Report

The presented manuscript is generally well written, concise and legible. The authors present an interesting study comparing the effects of transplantion of human MSCs from patients with osteoporosis and osteoarthritis in a mouse bone fracture model. The results are presented in a careful and convincing manner (small suggestions for changes are included in the comments in the PDF file). Additionally, the authors compare the effects of local and systemic transplantation. The results are interesting. In neither experimental group, MSC transplantation had a positive effect on the bone fracture healing (this information should be more emphasized in the manuscript), while in the "osteoarthritis" group, administration of MSC deteriorated the bone healing process in comparison to the control. The authors properly describe the limitations of their study. I believe the manuscript is suitable for publication. In my opinion, it requires minor corrections - all detailed comments and suggestions are in the attached PDF file.

Author Response

Point 1a: “Thus, if they are to be used for bone regeneration, they would likely be similarly valid to MSCs from non-osteoporotic individuals of similar age.” This sentence is nuclear and should be correct. What does it mean: “they would likely be similarly valid to MSCs from non-osteoporotic”? In the manuscript there is no data justifying such a tesis.

Response 1a: We have rephrased the text trying to make it easier to understand.

Point 1b:  On the other hand, there should be information that in this study actually non of MSCs groups did not cause improvement in comparison to the control. This seems to be of prime importance.

Response 1b: Thanks for the suggestion. We now comment on this in Conclusions.

Point 2: “patients” In the discusión you raise the issue of donor age and age associated changes in cell properties. Please, provide data about the age of donors used in this study.

Response 2: I have added that information.

Point 3: “osteoporotic hip fracture or osteoarthritis” Please add inofrmation, from how may patients the cells were used in the study. Were the cells from one group of patients pooled for transplantation or the populations were treated separately? Please, specify.

Response 3:  Only cells from a single patient were infused into each mouse. This is now stated in Methods. Therefore, the number of patients was equal to the number of mice shown in results.

Point 4: “patients with osteoporosis” As previously – specify from how may donors cells were used.

Response 4: See response to previous question.

Point 5: “model B was divided” Were cells in both groups injected throug the same needle? Please provide the information.

Response 5: Yes, cells were injected through the same needle in both groups, 25 G x 5/8” (0.50 x 16 mm).

Point 6: “(a) Group A3” Please, standardize the markings. Lowercase letters should be also in the figure (but larger font to make the figure more legible).

Response 6: It has been corrected in the figures.

Point 7: “Kruskal-Wallis and Mann Whitney U” The statistical methods should be better described. Why non-parametric test have been chosen?

Response 7: As the number of samples per group was small and the data departure from the normal distribution, non-parametric tests were the right statistical procedure to use.

Point 8: “BV/TV of the callus región” Not clear, why group “osteoporosis” is called “fracture” group here and later in the result section. Should be consistent in the whole manuscript.

Response 8: It has been modified. The correct name of the group is “osteoporosis”.

Point 9: “Scores of consolidation and corticalization in model A” Although I understand that in case of multiply comparisons it is not easy, but the authors could consider marking statistical significance on figures additionally (not only in the text). This would make the manuscript easier to read. Currently, the view of the figure itself does not say anything about the significance of the diffrences between groups.

Response 9: p-value has been added in the figures.

Point 10: “sistemic” should be “local”.

Response 10: It has been corrected.

Point 11: “systemic administration” should be “model B, local administration”.

Response 11: It has been corrected.

Point 12: “Thus, if they are to be used for bone regeneration, they would likely be similarly valid to MSCs from non-osteoporotic individuals of similar age.” The same as in abstract. This sentence is not clear. Do the authors suggest that MSC from helathy patients would have similar effect as MSC from osteoporotic patients? On what basis? The issue of donor age is also not clear. In this study, there is no comparison of effect in regards to the donor age. It is debatable if the issue of cell donor age should be mentioned in the conclusión section. On the other hand the should be be information that in this study actually non of MSCs groups did not cause improvement in comparison to the control. This semms to be of prime importance.

Response 12: Thanks for the comment. Some reports in the literature suggested that the functionality of MSCs decreases with the donor age.  We now include information about patients’ age. We have also modified the final sentence as explained above. It is to note patients with hip fractures were somewhat older than those with osteoarthritis. Despite that, the impact of their MSCs on fracture healing was not worse than that of patients with osteoarthritis.  

Point 13: “Supplementary Materials” If I undrestand well, these are actually the figures from the manuscript. I dont think they should be called “supplementary material”.

Response 13: Sorry for the mistake, this section has been removed.

Round 2

Reviewer 1 Report

Accept